# Raman Characterization of Carrier Concentrations of Al-implanted 4H-SiC with Low Carrier Concentration by Photo-Generated Carrier Effect

Tao Liu [1], Zongwei Xu [1,*], Mathias Rommel [2,*] , Hong Wang [3,*], Ying Song [1], Yufang Wang [4] and Fengzhou Fang [1]

1. State Key Laboratory of Precision Measuring Technology & Instruments,
   Centre of Micro/Nano Manufacturing Technology, Tianjin University, Tianjin 300072, China
2. Fraunhofer Institute for Integrated Systems and Device Technology IISB, Schottkystrasse 10,
   91058 Erlangen, Germany
3. State Key Laboratory of Separation Membranes and Membrane Processes,
   School of Materials Science and Engineering, Tianjin Polytechnic University, Tianjin 300387, China
4. School of Physics, Nankai University, Tianjin 300071, China
* Correspondence: zongweixu@tju.edu.cn (Z.X.); mathias.rommel@iisb.fraunhofer.de (M.R.);
  waho7808@163.com (H.W.)

**Abstract:** In this work, 4H SiC samples with a multilayer structure (shallow implanted layer in a lowly doped n-type epitaxial layer grown on a highly doped thick substrate) were investigated by Raman scattering. First, Raman depth profiling was performed to identify characteristic peaks for the different layers. Then, Raman scattering was used to characterize the carrier concentration of the samples. In contrast to the conventional Raman scattering measuring method of the Longitudinal Optical Plasmon Coupled (LOPC) mode, which is only suitable to characterize carrier concentrations in the range from $2 \times 10^{16}$ to $5 \times 10^{18}$ cm$^{-3}$, in this work, Raman scattering, which is based on exciting photons with an energy above the band gap of 4H-SiC, was used. The proposed method was evaluated and approved for different Al-implanted samples. It was found that with increasing laser power the Al-implanted layers lead to a consistent redshift of the LOPC Raman peak compared to the peak of the non-implanted layer, which might be explained by a consistent change in effective photo-generated carrier concentration. Besides, it could be demonstrated that the lower concentration limit of the conventional approach can be extended to a value of $5 \times 10^{15}$ cm$^{-3}$ with the approach presented here.

**Keywords:** Raman spectroscopy; silicon carbide; LOPC (Longitudinal Optical Plasmon Coupled) mode; carrier concentration; photo-generated carriers

## 1. Introduction

The third-generation semiconductor materials have superior physical and chemical properties, such as large forbidden band width (larger than 2.3 eV), high thermal conductivity, high carrier mobility and high breakdown voltage [1]. They can be widely used for devices operating under extreme conditions, such as high temperature, high voltage and high power [2]. The third-generation semiconductor materials primarily include silicon carbide, gallium nitride and diamond, etc. Silicon carbide (SiC) is known to have more than 200 polytypes [3], such as: 2H, 3C, 4H, 6H, 15R, which number refers to the total number of Si–C bilayers in the stacking sequence, and the letter refers to the crystal system (C for cubic, H for hexagonal and R for rhombohedral), as shown in Figure 1.

Although each polytype has the same chemical composition, it has different lattice structure and physical properties [4].

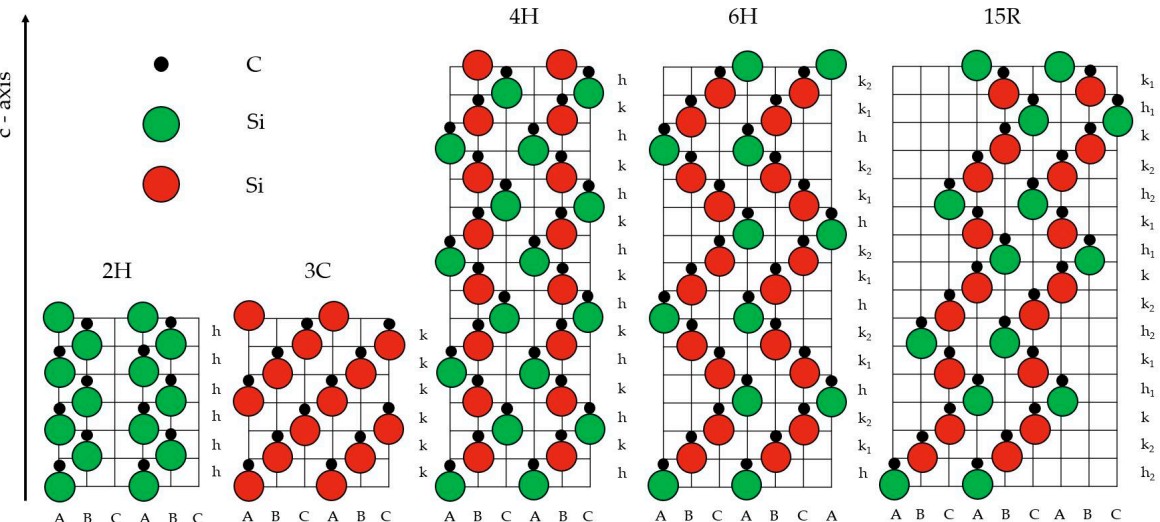

**Figure 1.** Diagram of the primitive cells of important polytypes of silicon carbide (SiC).

Several technologically-relevant silicon carbide polytypes, such as 4H and 6H, belong to the factor group $C_{6V}^4$. In the direction of the a-axis, 4H-SiC and 6H-SiC are almost similar. However, in the c-axis direction, the 4H-SiC polytype consists of four units, whereas 6H-SiC consists of six units [5]. 6H-SiC has been widely used in prototype devices such as thyristors and power metal oxide semiconductor field effect transistors (MOSFETs) [6,7]. Today 4H-SiC is dominating commercially-available SiC devices because of its higher electron mobility and weaker anisotropy in electron transport properties relative to 6H-SiC [8].

The carrier concentration is a key parameter of semiconductor devices and materials [9]. Accurate measurement of the carrier concentration, or at least the dopant concentration of 4H-SiC is extremely valuable [10,11]. Conventional methods for measuring the carrier or doping concentration of semiconductor materials include Hall effect measurements, C-V (Capacitance-Voltage) characterization or SIMS (secondary ion mass spectroscopy) characterization [12–15]. However, usually, time-consuming or destructive sample preparation is the disadvantage of the above methods. Raman spectroscopy is an optical characterization method that detects vibrations based on the inelastic scattering of light [16–18]. The Raman spectrum of a crystallized compound is composed of internal modes of vibration due to the polyatomic species and of external modes characterizing the lattice, thus related to the crystalline symmetry. For n-type 4H-SiC-doped samples, a fit of the theoretical formula of the LOPC (Longitudinal Optical Plasmon Coupled) mode peak in the Raman spectrum can be used to determine the carrier concentration non-destructively and rapidly [19–21]. However, the method is only applicable for carrier concentrations ranging from $2 \times 10^{16}$ to $5 \times 10^{18}$ cm$^{-3}$ [22].

The phenomenon of photo-generated carriers refers to the fact that when the energy of incident photons is equal to or larger than the forbidden band gap of the semiconductor, electrons in the valence band absorb the energy of the incident photons, transit into the conduction band and form electron-hole pairs simultaneously, as shown in Figure 2 [23]. The forbidden band gap for 4H-SiC is 3.24 eV, and it has been found in the experiment that when the surface of the 4H-SiC sample is irradiated with a 325 nm laser whose photon energy is 3.82 eV, the measured carrier concentration increases linearly with increasing laser power, which is primarily due to the contribution of photo-generated carriers [1].

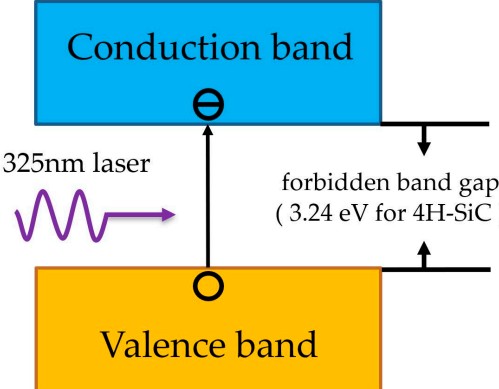

**Figure 2.** Schematic of 4H-SiC photo-generated carrier phenomenon.

According to the exciting wavelength with the transition between the conduction band and the valence band, lower concentrations of carriers are detected by Raman scattering. It will be shown that this modified Raman scattering measuring method extends the lowest measurement limit of the sample carrier concentration down to $5 \times 10^{15}$ cm$^{-3}$.

## 2. Materials and Methods

### 2.1. Raman Measurements

The Raman spectra were characterized both by a Horiba XploRA PLUS confocal Raman spectrometer and a Renishaw inVia laser micro-Raman spectrometer. The room temperature range during the Raman measurement was 25 ± 1 °C. The Renishaw inVia laser micro-Raman spectrometer (New Mills, UK) was equipped with a 30 mW 325 nm He–Cd laser to study the photo-generated carrier concentration of 4H-silicon carbide (SiC). The used objective lens was a 40× objective with NA = 0.50, the grating density was 3600 gr/mm, the spectrometer focal length is 250 mm, the size of the pixel of the detector is 26 μm. By changing the attenuator in the optical path, the laser power reaching the sample surface could be adjusted to 20 mW, 10 mW, 2 mW and 1 mW, respectively. Additionally, the Raman spectra were obtained in a backscattering geometry, z( - , - )z, where z is parallel to the c-axis of the crystal.

The Horiba XploRA PLUS confocal Raman spectrometer (Minami-ku, Kyoto, Japan) operated in a pinhole confocal mode, where the detected volume (depth range) of the analyzed sample can be changed by adjusting the diameter of the confocal pinhole (100 μm, 300 μm, 500 μm). The spectrometer is calibrated by a Si chip, and its focal length is 250 mm, the size of the pixel of the detector is 16 μm. It is equipped with a 10 mW 532 nm Nd: YAG SHG laser. The used objective lens was a 50× long distance objective lens with NA = 0.50, the grating density was 2400 gr/mm, and the spectral resolution was approximately 0.8 cm$^{-1}$. The smallest pinhole diameter (100 μm) was used in the experiment to measure the sample volume near the focus plane. The position of the focal plane in the depth direction was adjusted by the mechanical motor, whose spatial resolution in depth was approximately 0.1 μm. In this way, Raman signals could be obtained from different depths of the characterized sample.

### 2.2. Sample Description

The analyzed 4H-SiC samples have an epi layer with a relatively low doping concentration (i.e., $5 \times 10^{15}$ or $1.4 \times 10^{16}$ cm$^{-3}$) grown on commercial n-type heavily doped substrates. And the samples were cut 4° off the c-axis. The samples were further modified by Al or N ion implantation. More details on the preparation of samples Q1, Q4 and Q3 can be found elsewhere [24] and samples epi, A–N and A–P were prepared similarly. The thickness of the layer modified by ion implantation was approximately 300 nm. In order to reduce the lattice damage caused by the ion implantation process, and to electrically activate the implanted ions, the samples were then subjected to high-temperature

annealing treatment at 1700 °C for 30 min. The sample label and detailed information regarding the sample characteristics are listed in Table 1, whereas a schematic representation of the samples' depth profile is shown in Figure 3.

**Table 1.** The 4H-SiC samples information.

| Label | Implantation Parameter | | Annealing Conditions | | Epi Layer | |
| | Element | Surface Concentration (cm$^{-3}$) | Temperature (°C) | Time (min) | Thickness (µm) | Concentration (cm$^{-3}$) |
|---|---|---|---|---|---|---|
| epi | N | - | 1700 | 30 | - | $\approx 5.0 \times 10^{15}$ |
| Q1 | Al | $5.0 \times 10^{16}$ | 1700 | 30 | 5 | $\approx 5.0 \times 10^{15}$ |
| Q4 | Al | $5.0 \times 10^{17}$ | 1700 | 30 | 5 | $\approx 5.0 \times 10^{15}$ |
| Q3 | Al | $1.0 \times 10^{18}$ | 1700 | 30 | 5 | $\approx 5.0 \times 10^{15}$ |
| A-N | N | $5.0 \times 10^{19}$ | 1700 | 30 | 7.5 | $\approx 1.4 \times 10^{16}$ |
| A-P | Al | $5.0 \times 10^{19}$ | 1700 | 30 | 7.5 | $\approx 1.4 \times 10^{16}$ |

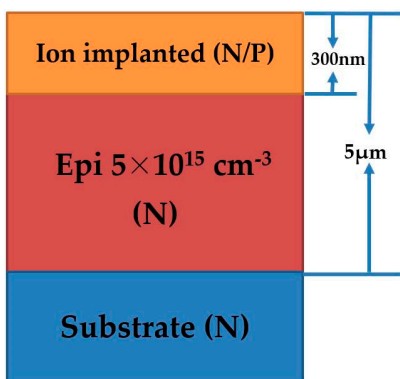

**Figure 3.** Schematic depth profile of 4H-SiC sample.

## 2.3. Determination of Carrier Concentration from Raman Measurements

Electro-optical and deformation potential mechanisms dominate the scattering process in the semiconductor with a wide band gap and low carrier mobility like silicon carbide [25]. Klein and his colleagues [26] demonstrate that the polarization selection rules are valid for both coupled and uncoupled longitudinal optical (LO) phonon mode. The LOPC mode is the result of the coupling between plasmon and LO phonons that interact through macroscopic electric fields [27]. The theoretical analysis of the coupled mode in the cubic crystal had been extended to the hexagonal system [10,28,29]. The Raman intensity of the LOPC mode can be written by [22,30]:

$$I(\omega) = SA(\omega)\mathrm{Im}\left(\frac{-1}{\varepsilon(\omega)}\right) \tag{1}$$

where $\omega$ is the Raman shift, S is the proportionality parameter, $\varepsilon(\omega)$ is the classical dielectric function. $A(\omega)$ is expressed by:

$$A(\omega) = 1 + \frac{2C\omega_T^2\left[\omega_P^2\gamma(\omega_T^2-\omega^2)-\omega^2\Gamma_T(\omega^2+\gamma^2-\omega_P^2)\right]}{\Delta} + \left(\frac{C^2\omega_T^4}{\Delta}\right) \times$$
$$\left\{\omega_P^2\left[\gamma\left(\omega_L^2-\omega_T^2\right)+\Gamma_T\left(\omega_P^2-2\omega^2\right)\right]+\omega^2\Gamma_T\left(\omega^2+\gamma^2\right)\right\}/\left(\omega_L^2-\omega_T^2\right) \tag{2}$$

where $\omega_T$ is the vibration frequency of the uncoupled transverse optical phonon mode, $\omega_L$ is the vibration frequency of uncoupled longitudinal optical phonon mode, $\omega_P$ is the vibration frequency

of plasmons, $\gamma$ is the plasmon attenuation constant, $\Gamma_T$ is the attenuation constant of the transverse optical phonons. $\Delta$ is given by:

$$\Delta = \omega_P^2 \gamma \left[ \left( \omega_T^2 - \omega^2 \right)^2 + (\omega \Gamma_T)^2 \right] + \omega^2 \Gamma_T \left( \omega_L^2 - \omega_T^2 \right) \left( \omega^2 + \gamma^2 \right) \tag{3}$$

C is the Faust-Henry coefficient, which is related to the ratio of the Raman intensity of the longitudinal optical phonon mode to that of the transverse optical phonon mode in the pure silicon carbide crystal [30].

$$\frac{I_{LO}}{I_{TO}} = \left( \frac{\omega_1 - \omega_L}{\omega_1 - \omega_T} \right)^4 \frac{\omega_T}{\omega_L} \left( 1 + \frac{\omega_T^2 - \omega_L^2}{C\omega_T^2} \right)^2 \tag{4}$$

where $\omega_1$ represents the frequency of the incident light. The classical dielectric function is obtained by the combination of phonons and plasmons:

$$\varepsilon(\omega) = \varepsilon_\infty \left( 1 + \frac{\omega_L^2 - \omega_T^2}{\omega_T^2 - \omega^2 - i\omega \Gamma_T} - \frac{\omega_P^2}{\omega(\omega + i\gamma)} \right) \tag{5}$$

Here, $\varepsilon_\infty$ is the optical dielectric constant, and i is the imaginary number. The plasmon frequency is given by Equation (6):

$$\omega_P = \left( \frac{4\pi n e^2}{\varepsilon_\infty m^*} \right)^{\frac{1}{2}} \tag{6}$$

Here, $n$ is the carrier concentration, $e$ is the elementary charge, and $m^*$ is the electron effective mass. Harima et al. [22] and Burton et al. [31] verified that the Raman scattering measuring method is applicable to determine the carrier concentration for n-type doped samples. The suitable carrier concentration for this method is in the range from $2 \times 10^{16}$ to $5 \times 10^{18}$ cm$^{-3}$, but for 4H-SiC with low carrier concentration, the calculation error of the method is rather large [8,32]. Additionally, when the carrier concentration is in the range of $2 \times 10^{16}$ cm$^{-3}$ to $3 \times 10^{18}$ cm$^{-3}$, the carrier concentration has a linear relationship with the Raman peak position of the LOPC mode [33].

The experimental data was fitted to the theoretical equation by mathematical analysis software MATLAB R2018a. For fitting, the parameters S, $\Gamma$, $\gamma$, $\omega_P$ were used as adjustable parameters, and their numerical values were obtained by the MATLAB fitting procedure. Finally, the value of the carrier concentration was obtained by Equation (6). Table 2 shows selected material constants values to fit the experimental Raman spectroscopy data.

**Table 2.** The material constants used to fit experimental Raman spectroscopy [22].

| LO Phonon Frequency $\omega_L$ (cm$^{-1}$) | TO Phonon Frequency $\omega_T$ (cm$^{-1}$) | Faust-Henry Coefficient $C$ | Optical Permittivity $\varepsilon_\infty$ | Carrier Effective Mass $m^*$ (kg) |
|---|---|---|---|---|
| 964.2 | 777.0 | 0.43 | 6.78 | 0.48m$_0$ (m$_0$ = 9.11 × 10$^{-31}$ kg) |

## 3. Results

As the investigated samples have a non-homogeneous dopant, thus, carrier concentration depth profile confocal Raman spectroscopy was adopted for the depth profiling and analysis. In the first Section 3.1, carrier concentrations in the substrate layer will be determined. For that purpose, depth-dependent Raman measurements were performed to determine optimum measurement conditions and parameters. Then, the photo-generated carriers method was employed to extend the lowest applicable range of the Raman spectroscopy measuring method for 4H-SiC (see Section 3.2).

### 3.1. Depth Analysis and Results

Figure 4 shows Raman spectra of the Q1 sample for different laser powers using the 532 nm laser. The abscissa and ordinate of the figure were the position and relative intensity of the Raman peaks, respectively. One can detect three Raman peaks, one at approximately 777 cm$^{-1}$ which is related to the TO peak (see Figure 4a), one at 964.2 cm$^{-1}$ which is related to the LO mode, and one at approximately 980 cm$^{-1}$, which is related to the LOPC mode (see Figure 4b). It was considered that the two peaks result from the double-layer characteristic of the sample, that is, the double peaks contain the Raman information of the surface epitaxial layer and the layer beneath, i.e. the substrate.

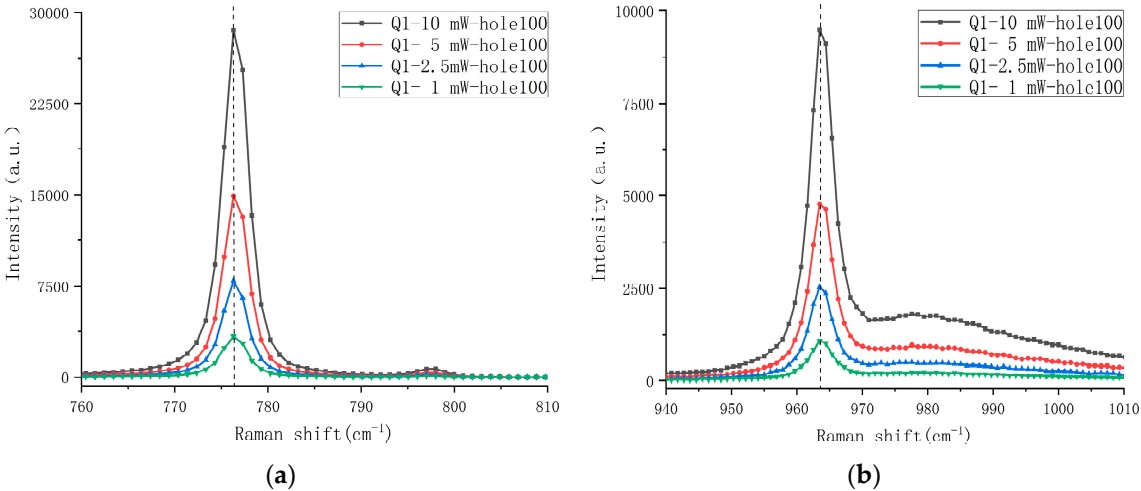

**Figure 4.** Raman spectra of the Q1 sample measured with exciting line at 532 nm with the 100 μm pinhole, (**a**) TO mode, (**b**) LO and Longitudinal Optical Plasmon Coupled (LOPC) modes, using different laser powers as indicated in the legend, whilst focusing on the sample surface.

In order to verify the assumption mentioned above, Raman depth profiling was performed, as shown in Figure 5.

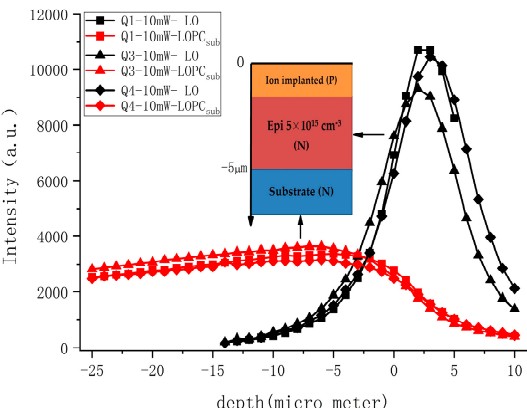

**Figure 5.** Relative Raman intensity of LO and LOPC$_{sub}$ peaks of Q1, Q3 and Q4 samples with exciting line at 532 nm with the 100 μm pinhole at different focal plane depths.

Figure 5 shows Raman information of Q1, Q3 and Q4 samples for different depths. The abscissa represents the position of the focal plane, where negative values refer to focal planes inside the sample, and the ordinate gives the relative intensity of the Raman peaks. The black line represents the intensity of the LO peak, and the red line represents the intensity of the LOPC peak. When the focal plane raises from −25 to −5 μm, the intensity of the LOPC peak is almost constant or only slightly increasing.

However, when the focal plane position raises from −5 to +5 μm, the intensity of the LOPC peak decreases rapidly. Conversely, the intensity of the LO peak increases rapidly when the focal plane position raises from −5 to 0 μm. As the epi layer thickness is 5 μm, it is assumed that the LOPC peak in the depth profiling reflects the Raman information of the heavily-doped substrate layer and labeled as LOPC$_{sub}$ (as indicated in Figure 5), while the LO peak gives the Raman information of the epi layer with low carrier concentration.

Therefore, it is suggested that the presented Raman depth analysis is an efficient method to characterize samples with a multilayer structure, as the above experimental results are highly consistent with the sample's structure. By numerically fitting the separated LOPC$_{sub}$ peak experimental data with focal plane at −25 μm, the calculated carrier concentrations of the heavily-doped substrate layer by Equation (6) were in the range from $3.9 \times 10^{18}$ to $4.1 \times 10^{18}$ cm$^{-3}$, which is highly consistent with the concentrations of commercially-available 4H-SiC substrates.

However, when trying to apply the same evaluation procedure to determine the carrier concentration of the epi layer, no reasonable carrier concentrations could be obtained because of the very low concentration of the epi layer, which is below the applicable range of the method as mentioned before. To overcome this limitation, a modified approach is introduced in the next section.

### 3.2. Experimental Raman Results with Photo-Generated Carriers

The energy of a photon with a 325 nm wavelength is 3.82 eV, while for the exciting line at 532 nm it is 2.33 eV. The forbidden band gap of 4H-SiC is 3.24 eV, so 325 nm laser photons can generate photo-generated electron-hole pairs, but 532 nm will not. Coupled with photo-generated carrier plasmon, LO mode changes to LOPC mode. Therefore, it is labeled as LOPC$_{pgc}$ in the following, to be distinguished from LOPC$_{sub}$.

The absorption depth of the exciting line at 325 nm in SiC is approximately 4 μm, which has been confirmed by Xu et al. and Derst et al. [34,35]. With such laser, the LOPC$_{sub}$ peak could not have been found in any Raman spectrum of the investigated epi sample, because the epi layer thickness of the tested samples is larger than 5 μm, which supports the above conclusion that the LOPC$_{sub}$ peak shown in Figure 5 is originating from the substrate only. The Raman spectra of the epi samples with this exciting line at 325 nm using different laser power are exhibited in Figure 6. The detailed relationship between Raman peak position of the TO & LOPC$_{pgc}$ modes of epi samples and the laser power is shown in Figure 7.

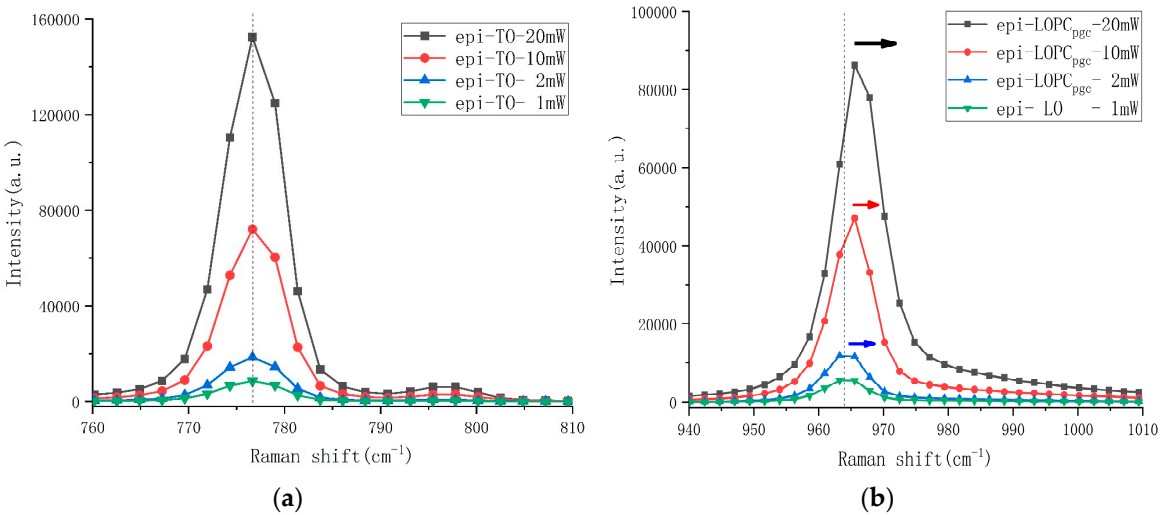

**Figure 6.** Raman spectra of the epi sample with exciting line at 325 nm, (**a**) TO modes, (**b**) LOPC$_{pgc}$ modes, using different laser power (as indicated in the legend).

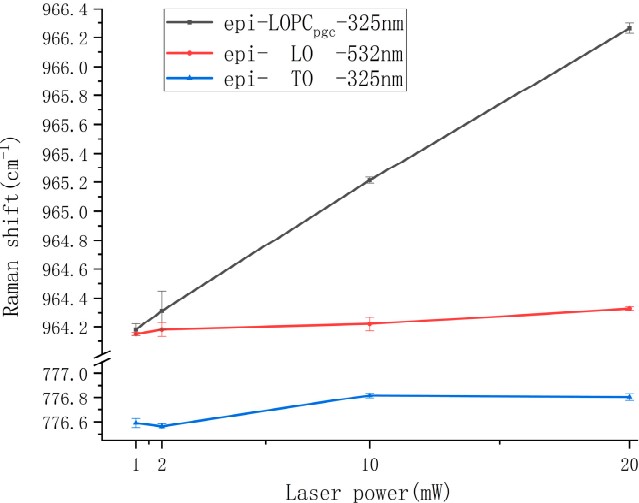

**Figure 7.** Dependence of Raman peak position of the TO & LOPC$_{pgc}$ modes for epi samples on laser power for exciting line at 325 and 532 nm.

N. Piluso et al. [23] stated that for an incident laser power of 1 mW, no increase in carrier concentration could be detected, because it was below the sensitivity of their Raman tools. The Raman scattering measurements with 325 and 532 nm showed that the peak position of the LOPC$_{pgc}$ mode measured by the 325 nm laser at 1 mW was the same as the LO peak position measured at 532 nm without the photo-generated carrier phenomenon (see Figure 7), which supports the above statement. Therefore, in Figure 6b, the LOPC$_{pgc}$ label changes back to LO when the laser power is 1 mW.

The specific information of the Raman peak positions for measurements with the 325 nm laser are shown in Table 3 for three measurement positions on the sample. When measuring with the exciting line at 325 nm, the peak position of the LOPC$_{pgc}$ mode is linearly correlated with the laser power, while the Raman shift of the TO peak is almost constant. However, when measuring with the exciting line at 532 nm, there is no change of the Raman peak position both for the LOPC$_{pgc}$ peak and TO peak with increasing laser power, as shown in Figures 4 and 7.

**Table 3.** Peak position of the Raman LOPC$_{pgc}$ and TO peaks for the epi sample at three different measurement positions when using the exciting line at 325 nm.

| Laser Power (mW) | epi-1-LOPC$_{pgc}$ Position (cm$^{-1}$) | epi-2- LOPC$_{pgc}$ Position (cm$^{-1}$) | epi-3- LOPC$_{pgc}$ Position (cm$^{-1}$) | Average Value (cm$^{-1}$) |
|---|---|---|---|---|
| 1 | 964.22 | 964.18 | 964.14 | 964.18 |
| 2 | 964.25 | 964.27 | 964.47 | 964.31 |
| 10 | 965.23 | 965.22 | 965.19 | 965.21 |
| 20 | 966.30 | 966.26 | 966.23 | 966.26 |
| **Laser Power (mW)** | **epi-1-TO Position (cm$^{-1}$)** | **epi-2-TO Position (cm$^{-1}$)** | **epi-3-TO Position (cm$^{-1}$)** | **Average Value (cm$^{-1}$)** |
| 1 | 776.62 | 776.60 | 776.55 | 776.59 |
| 2 | 776.57 | 776.58 | 776.54 | 776.56 |
| 10 | 776.81 | 776.84 | 776.80 | 776.82 |
| 20 | 776.78 | 776.84 | 776.79 | 776.80 |

By numerically fitting the LOPC$_{pgc}$ peak curve when using the exciting line at 325 nm, the carrier concentrations under different laser power were calculated by Equation (6). Then, as shown in Figure 8,

a linear relationship between the carrier concentration of the epi sample and the laser power was found which can be expressed by Equation (7).

$$n = 7.37E15 \text{ cm}^{-3}/\text{mW} * p - 1.49E15 \text{ cm}^{-3} \tag{7}$$

where n is the carrier concentration and p is the laser power (in mW). When the laser power p is set to 1 mW, the carrier concentration of the epi sample is calculated as $5.9 \times 10^{15}$ cm$^{-3}$. The calculated result is close to the carrier concentration $5.0 \times 10^{15}$ cm$^{-3}$ measured by C-V testing, with about 20% deviation between the two methods. Here defining a new parameter K to represent the capability of the sample photo-generated carrier,

$$K = \frac{n_1 - n_0}{p_1 - p_0} \tag{8}$$

where $n_1$ is the carrier concentration with laser irradiation, $n_0$ is the carrier concentration without laser, $p_1$ is the incident laser power and $p_0$ equals to 1 mW. For instance, the parameter K value of the epi sample is $7.37 \times 10^{15}$ cm$^{-3}$/mW, which means that for each 1 mW increase in laser power, the photo-generated carrier concentration increases by approximately $7.37 \times 10^{15}$ cm$^{-3}$.

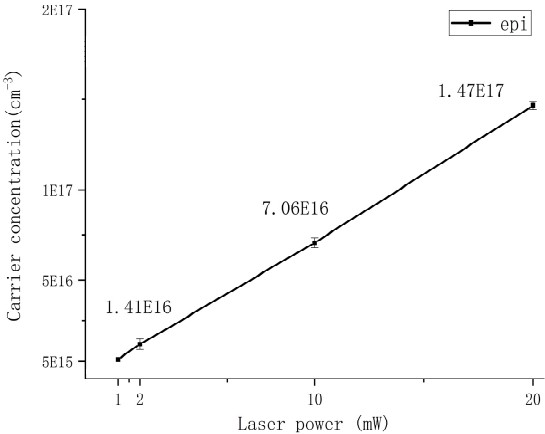

**Figure 8.** Linear relationship of carrier concentration of the 4H-SiC epi sample with laser power under the exciting line at 325 nm.

From literature it is known that for n-type 4H-SiC, the LOPC mode Raman peak shifts to larger wavenumbers and becomes broadened with increasing carrier concentration [22,27,32]. For p-type, though, the relationship between the LOPC peak and carrier concentration is significantly different. Mianti [36] researched the Raman spectra of p-type 4H-SiC, which showed that the LOPC peak shifted towards smaller wavenumbers and broadened with increasing carrier concentration. Since the LOPC peak of the implanted samples investigated here contains the Raman information both from the epi layer and the implanted layer, the value of the parameter K cannot be fitted precisely. However, the parameter $K_1$ can be similarly defined to represent the effect of the modified layer on the photo-generated carrier capacity of the sample qualitatively,

$$K_1 = \frac{\omega_{PG} - \omega_{LO}}{p_1 - p_0} \tag{9}$$

where $\omega_{PG}$ is the LOPC$_{pgc}$ Raman peak position with laser irradiation, $\omega_{LO}$ is the LO Raman peak position, $p_1$ is the incident laser power and $p_0$ equals to 1 mW. The fitted value of $K_1$ parameter for other samples is shown in Table 4.

**Table 4.** $K_1$ value for tested samples with exciting line at 325 nm.

| Label | epi | Q1 | Q4 | A-N | A-P |
|---|---|---|---|---|---|
| $K_1$ (cm$^{-1}$/mW) | 0.11 | 0.10 | 0.08 | 0.12 | 0.06 |

The relationship between the LOPC$_{pgc}$ mode Raman peak position of the samples with different ion-modified layer and different laser power is shown in Figure 9, and the dotted lines represent the linear fit results for $K_1$. When the laser power increases from 1 to 20 mW, the LOPC$_{pgc}$ mode Raman peak position of the epi sample without any ion implantation shifts towards larger wavenumbers by approximately 2 cm$^{-1}$, and the $K_1$ value is 0.11 cm$^{-1}$/mW for the epi sample. However, for the Al ion-implanted Q series samples, the phenomenon of the Raman peak shifting towards larger wavenumbers with increasing laser power gradually weakens with the increase of aluminum ion concentration in the modified layer, and the $K_1$ values for the Q series samples decrease from 0.10 to 0.08 cm$^{-1}$/mW with increasing Al ion-implanted concentration, and $K_1$ becomes 0.06 cm$^{-1}$/mW for the highest Al concentration in the sample A-P.

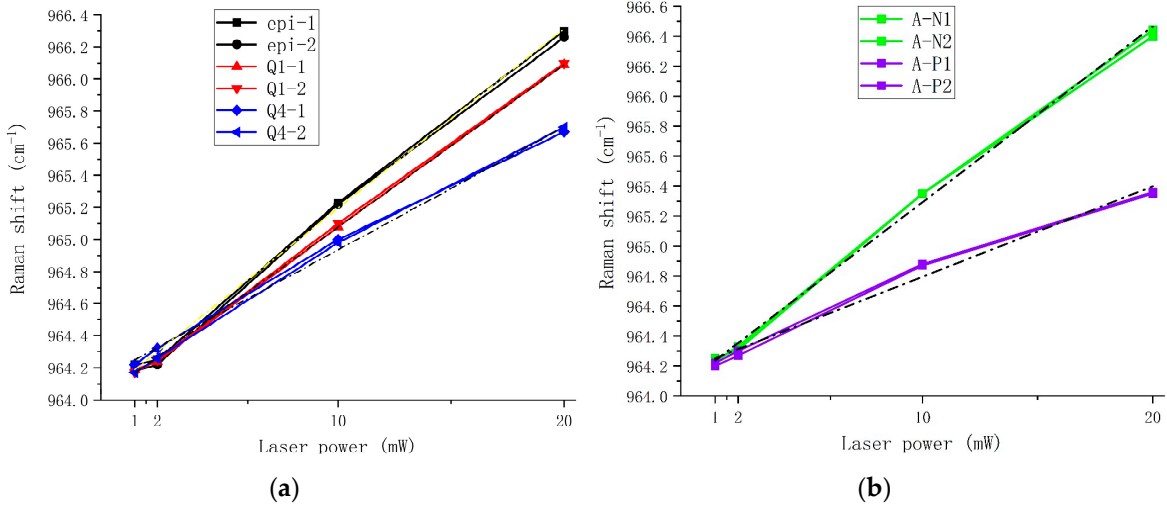

**Figure 9.** LOPC$_{pgc}$ mode Raman peak position for different samples as a function of laser power: (**a**) For epi and Q series samples, (**b**) for A-N and A-P samples. Labels 1&2 mean two measurements at different positions on the same measured sample.

Conversely, for the N ion-implanted A-N sample, the phenomenon of the laser-increasing Raman peak shifting becomes strengthened, where $K_1$ is increased to 0.12 cm$^{-1}$/mW. When the laser power is increased to 20 mW, the LOPC$_{pgc}$ peak position of the A-N samples is approximately 1.0 cm$^{-1}$ above the peak position of the A-P samples, as shown in Figure 9b.

Overall, by analyzing the value of $K_1$ for different samples, the effect of the ion-modified layer on photo-generated carriers in 4H-SiC can be explained consistently with the trends known from literature. As a result of the neutralization of free electrons and holes that derived from the p-type ion-implanted layer, the overall concentration of effective photo-generated carriers is reduced to some extent. On the other hand, Al ion implantation causes defects in the crystal lattice, like the carbon or silicon vacancies [37,38], etc. The higher the implantation dose, the more serious the lattice damage. There are more defects in the high implant dose sample, so it also has a negative impact on the generation of photo-generated carriers.

## 4. Conclusions

In this work, Raman depth profiling was successfully applied to determine the different Raman properties from different layers of 4H-SiC samples with a multilayer structure (shallow implanted layer

in a lowly doped n-type epitaxial layer grown on a highly doped thick substrate). Most importantly, it could be shown that Raman scattering measurements with exciting photons having an energy above the band gap of 4H-SiC can be used to determine carrier concentrations of rather lowly doped samples. The approach is based upon a linear relationship between the effective free carrier concentration and the laser power from which the carrier concentration of the n-type epitaxial layer could be determined to be $5.9 \times 10^{15}$ cm$^{-3}$, which is very close to the actual doping concentration of $5.0 \times 10^{15}$ cm$^{-3}$ measured by C-V testing. Hence, the presented Raman approach extends the lowest measurable carrier concentration down to $5 \times 10^{15}$ cm$^{-3}$, which is approximately four times more sensitive compared to former approaches. For the presented approach, the shift of the peak position of the LOPC$_{pgc}$ mode with increasing laser power is mainly evaluated and described by the parameter K$_1$. As a result of the neutralization of free electrons and holes derived from an Al-implanted layer and defects in the crystal lattice caused by ion implantation, the value of K$_1$ decreases from 0.11 cm$^{-1}$/mW for the epi-layer to 0.06 cm$^{-1}$/mW with increasing Al-implanted concentration. This shows that the approach is even able to detect the influence of a very shallow surface near the Al-implanted layer on the effective carrier concentration.

**Author Contributions:** T.L. performed all the experiments, conceived and designed the layout; Z.X., M.R. and H.W. collected and contributed materials; Z.X., M.R., Y.S., Y.W. and F.F. provided valuable suggestions about the paper; T.L., Z.X. and Y.S. wrote the paper.

**Funding:** The study is supported by National Natural Science Foundation of China (No. 51575389, 51761135106), National Key Research and Development Program of China (2016YFB1102203), State key laboratory of precision measuring technology and instruments (Pilt1705), and the '111' project by the State Administration of Foreign Experts Affairs and the Ministry of Education of China (Grant No. B07014).

**Acknowledgments:** Authors thank Tao Xue from Tianjin University for valuable discussions.

**Conflicts of Interest:** The authors declare no conflict of interest.

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
