# Peer review of "Raman Characterization of Carrier Concentrations of Al-implanted 4H-SiC with Low Carrier Concentration by Photo-Generated Carrier Effect"

_crystals, doi:10.3390/cryst9080428_

Round 1
Reviewer 1 Report
The proposal of the article untitled "Raman characterization of carrier concentrations of Al implanted 4H-SiC with Low Carrier Concentration by Photogenerated Carrier Effect" by T. Liu et al is not publishable, the authors do not know enough Raman effect.
Page 1:
- Key words: correcting spelling mistakes,
- Introduction: line 9: C6v is not a crystalline symmetry that is characterized by its space group (Hermann-Mauguin notation) or its factor group (Schoenflies notation with mention of order)
Page 2 :
- introduction: lines 10-12: the Raman spectrum of a crystallized compound is composed of internal modes of vibration due to the polyatomic species and of external modes characterizing the lattice thus related to the crystalline symmetry.
- Materials and methods: How do you define spectral resolution? It is not enough to make a linear interpretation according to the number of groves/mm of the grating equipping different devices. Other factors contribute to the spectral resolution, such as the exciting wavelength, the focal length of the spectrometer, the size of the pixel of the detector etc ....
Moreover, the writings are in contradiction: FIGS. 3 and 5 show for the same spectral range, that fewer measurement points are obtained by exciting at 325 nm: which means that the spectral resolution is less good (higher value) than by exciting at 532 nm.
For the proposed work it is necessary to have a high spectral resolution of the spectrometer, the accuracy is not sufficient at 325 nm.
Page 3:
- line 1: What does "background geometry with the angle between the scattered light and the incident light vector being close to 180°" mean?
- Line 5: the ruby laser has no line at 532 nm
Page 5:
- Depth analysis and results line 2: the intensity of the lines is dependent on the polarization (see J. Raman Spectrosc 2009, 40, 1867-1874 or DOI 10.1002 / day.2334)
- Figure 3: How the authors explain that the intensity of the TO mode line is more intense than that of the LO mode when the recordings are obtained in a backscattering configuration?
Figures 3 and 5 need to be revised
Reference 34 needs to be revised
The argumentation by the authors is not correct, the precision on the position of the lines is weaker using λexc. = 325 nm and the lack of spectral resolution can lead to misinterpretation of the feature.
This article cannot be selected for publication in Crystal, I refuse it.

Reviewer 2 Report
Abstract
Second to the last sentence change “left shift” to shifts to lower wavenumber, or is redshifted. You need to be precise here in what you mean by left shifted
Introduction
If you are going to introduce polytypes you should probably explain what 2H, 3C, 4H, 6H and 15R are, meaning that the number refers to the number of layers in a repeat that occur in the stacking direction, and the letter refers to the crystal system (described by Ramsdell 1947 Studies on Silicon Carbide American Mineralogist, 32). A diagram showing the SiC structure could be added here.
What do you mean by in the direction of the lattice a? Do you mean along the a-axis? Do not confuse “lattice” with “structure” for polytyptes 4H and 6H they are very similar along the a-axis and differ in the length of the c-axis which is along the stacking direction. Again a crystal structure diagram would help out here
Page 2 last sentence of first paragraph replace “than” with “relative to” to read “… and weaker anisotropy in electron transport properties relative to 6H-SiC [7].”
Page 2 third paragraph the last sentence is there a reference for this?
Page 2 4th paragraph 1st sentence change measuring to measurement to read “…a modified Raman scattering measurement technique is proposed…”
Materials and methods
What is the spectrometer focal length of the Raman spectrometers that were used in this study?
How was the spectrometer calibrated? Si chip, Ne lamp?
How was the laser power that reached the sample measured?
Sample description, are these samples oriented?
Page 4 first paragraph Klein and his colleagues? Is this a reference?
Page 4 near the end, Harima et al. and Burton et al. add the number of these references to the text
Figure 4 you should probably label the inset of the sample configuration with the depth so that readers know where 0 is, meaning is 0 between the substrate and the Epi 5x1015 cm-1 layers?
Page 6 paragraph 2 what are the estimations on the uncertainty in these carrier concentration ranges that you find by equation 6?
Page 7 N. Piluso et al. add the number of this reference to the text
Do the authors think that there is any heating of the sample occurring under 325 nm excitation? I ask this because Bauer et al. (2009) report that for doped nitrogen 4H-SiC under heating from 3-112 oC with measurements takes parallel to the c-axis the 999.2 cm-1 mode shifts positively. I am just wondering if the authors have considered sample orientation and possible heating by the excitation source in these measurements?
Page 8 near the bottom, so is 10% the estimated uncertainty in this new method for measuring low carrier concentrations?
Page 9, the authors state that the phenomenon of laser increasing Raman peak shifting weakens as Al concentration increases, why is this? How does an increase in Al concentration inhibit this peak shifting? Also, I think this sentence needed reworded, “However, for the Al ion implanted Q series samples, the phenomenon of raman peak shifting to the right with an increase in laser power gradually weakens…” or something like that.
Round 2
Reviewer 1 Report
I have carefully read the corrections that are not all acceptable:
The answer to some question by references do not show that you understood the meaning of the question on the one hand and the writings in the references on the other hand.
Page 1, last line: C4v4 is not a space group but a factor group,
Page 2, line 7 of the 1st paragraph: the term "lattice" is to be deleted: the glasses have no lattice and have a Raman spectrum.
Page 2, last paragraph: This is not a new Raman scattering technique but the coupling with another technique that leads to new interpretations. The use of an excitation line at 325 nm makes it possible to work in pre-resonance mode and to detect lower concentrations of carriers. (see Resonance Raman effect and SERS effect).
Page 3, lines 8-9: delete the sentence with the laser polarization
Page 3, paragraph 2.2, line 8 replace "samples label" by "sample label"
Page 8, Fig. 6: By exciting at 325 nm, the lack of spectral resolution leads to an overlap of the two LO-and LOPC mode bands at 964 cm-1 and 980 cm-1, to give a single broad asymmetric band whose extremum shifts to high wavenumbers by increasing laser power. This slight evolution of the spectral feature is related to the increase of carriers with the laser power, resulting in an increase of the band at 980 cm-1. Deconvolution is then necessary to determine the contribution of each of these bands and then to realise acceptable measurements.
How can authors validate a variation of the band extremum of 2 cm-1, when the detector pixel receives on average more than 2 cm-1?
The conclusion must be rewritten: there are no old or new Raman scattering methods.
To be published, all this different points must be clarified (see also annotations on the pdf document).
